# A Pilot Clinical Study on Post-Operative Recurrence Provides Biological Clues for a Role of *Candida* Yeasts and Fluconazole in Crohn’s Disease

**DOI:** 10.3390/jof7050324

**Published:** 2021-04-22

**Authors:** Boualem Sendid, Nicolas Salvetat, Helène Sarter, Severine Loridant, Catherine Cunisse, Nadine François, Rachid Aijjou, Patrick Gelé, Jordan Leroy, Dominique Deplanque, Samir Jawhara, Dinah Weissmann, Pierre Desreumaux, Corinne Gower-Rousseau, Jean Frédéric Colombel, Daniel Poulain

**Affiliations:** 1INSERM U1285, CNRS UMR 8576, Glycobiology in Fungal Pathogenesis and Clinical Applications, Université de Lille, F-59000 Lille, France; boualem.sendid@univ-lille.fr (B.S.); severine.loridant@chru-lille.fr (S.L.); nadine.francois@chru-lille.fr (N.F.); rachid.aijjou@chru-lille.fr (R.A.); jordan.leroy@chru-lille.fr (J.L.); samir.jawhara@inserm.fr (S.J.); 2Pôle de Biologie—Pathologie—Génétique, Institut de Microbiologie, Service de Parasitologie Mycologie, CHU Lille, F-59000 Lille, France; 3ALCEDIAG Sys2Diag/CNRS UMR 9005 Parc Euromédecine Cap Delta, 1682 Rue de la Valsière, CEDEX 4, 34184 Montpellier, France; nicolas.salvetat@sys2diag.cnrs.fr (N.S.); dweissmann@alcediag-alcen.com (D.W.); 4U1286—INFINITE—Institute for Translational Research in Inflammation, INSERM, Université de Lille, F-59000 Lille, France; helene.sarter@chru-lille.fr (H.S.); pdesreumaux@hotmail.com (P.D.); corinne.gower@chru-lille.fr (C.G.-R.); 5CHU Lille, Unité de Biostatistiques, Pôle de Santé Publique, F-59000 Lille, France; catherine.cunisse@chru-lille.fr; 6Biological Resources Centre, Lille University Hospital, F-59000 Lille, France; patrick.gele@chru-lille.fr (P.G.); dominique.deplanque@chru-lille.fr (D.D.); 7INSERM, CHU Lille, CIC 1403—Centre d’Investigation Clinique, Université de Lille, F-59000 Lille, France; 8CHU Lille, Service d’Hépato-Gastroentérologie, F-59000 Lille, France; jean-frederic.colombel@mssm.edu; 9Department of Gastroenterology, Icahn School of Medicine at Mount Sinai, New York, NY 10029, USA

**Keywords:** *Candida*, Crohn’s, immune sensing

## Abstract

Background and aims: This study prompted by growing evidence of the relationship between the yeast *Candida albicans* and Crohn’s disease (CD) was intended to assess the effect of a 6-month course of the antifungal fluconazole (FCZ) on post-operative recurrence of CD. Methods: Mycological samples (mouth swabs and stools) and serum samples were collected from 28 CD patients randomized to receive either FCZ (*n* = 14) or placebo (*n* = 14) before surgical resection. Serological analysis focused on levels of calprotectin, anti-glycan antibodies, and antibody markers of *C. albicans* pathogenic transition. Levels of galectin-3 and mannose binding lectin (MBL) involved in *C. albicans* sensing and inflammation were also measured. Results: 1, 2, 3, and 6 months after surgery, endoscopy revealed recurrence in 5/12 (41.7%) patients in the FCZ group and 5/9 (55.6%) in the placebo group, the small cohort preventing any clinical conclusions. In both groups, surgery was followed by a marked decrease in *C. albicans* colonization and biomarkers of *C. albicans* pathogenic transition decreased to non-significant levels. Anti-glycan antibodies also decreased but remained significant for CD. Galectin-3 and calprotectin also decreased. Conversely, MBL levels, which inversely correlated with anti-*C. albicans* antibodies before surgery, remained stable. Building biostatistical multivariate models to analyze he changes in antibody and lectin levels revealed a significant relationship between *C. albicans* and CD. Conclusion: Several combinations of biomarkers of adaptive and innate immunity targeting *C. albicans* were predictive of CD recurrence after surgery, with area under the curves (AUCs) as high as 0.86. FCZ had a positive effect on biomarkers evolution. ClinicalTrials.gov ID: NCT02997059, 19 December 2016. University Hospital Lille, Ministry of Health, France. Effect of Fluconazole on the Levels of Anti-*Saccharomyces cerevisiae* Antibodies (ASCA) After Surgical Resection for Crohn’s Disease. Multicenter, Randomized, and Controlled in Two Parallel Groups Versus Placebo.

## 1. Introduction

The yeast *Candida albicans* is a frequent commensal of humans that can colonize all segments of the digestive tract and vagina. Its role in intestinal homeostasis remains largely unknown in contrast to its deleterious effects, which are well-documented. These range from mucocutaneous candidiasis, with vaginal infections occurring in 75% of women at least once in their lifetime, to systemic infections originating in the gut of hospitalized patients. The main risk factors are immunodeficiency related to the primary pathology and the medico-surgical procedures used to manage it, including broad-spectrum antibiotics [1]. In these patients, *C. albicans* undergoes saprophytic-pathogenic transition. *Candida* species now rank fourth among the causes of nosocomial infections with candidemia associated with mortality rates as high as 40–60% [2]. Considerable investment in basic research and the development of new antifungal drugs have not resulted in effective control of these life-threatening yeast infections.

The interest of the gastroenterology community started with the yeast *Saccharomyces cerevisiae*, ref. [3] a harmless yeast used in the food industry, and a popular model for cell and molecular biologists [4]. Medical interest came from the development, in our laboratory, of an ELISA test to detect human antibodies against mannan (a complex mannose polymer of the cell wall surface) that were present in a large proportion of patients with Crohn’s disease (CD). We named these antibodies “anti-*S. cerevisiae* antibodies” (ASCA), ref. [5] and their detection, combined with perinuclear anti-neutrophil cytoplasmic antibodies (pANCA), enabled the differentiation of CD from ulcerative colitis (UC) [6]. Further studies showed that ASCA were strong markers of CD and were present in up to 30% of first-degree relatives (FDRs) of CD patients [5] vs. 6% of controls [7]. These antibodies remain a potent serological marker of CD, and their presence predicts the onset of the disease [8,9]. From a pathophysiological point of view, early studies with conventional mycological methods established a link between ASCA and *C. albicans* by showing a correlation between ASCA and *C. albicans* colonisation in FDRs [10]. It was also shown that modulation of mannan epitopes of *C. albicans*, when growing in human tissues, ref. [11] was responsible for ASCA generation. This relationship was not limited to *C. albicans* mannan but extended to other important immunologically active components of the *C. albicans* cell wall, such as glucans and chitin. Combined serological and mycological studies in patients with CD or invasive *Candida* infection (ICI) showed that anti-laminaribioside (ALCA) and anti-chitobioside (ACCA), respectively, described as CD biomarkers, ref. [12] could also be produced during ICI [13]. The development of experimental models to study *C. albicans*-gut interactions confirmed that Dextran Sodium Sulfate (DS)induced inflammation in mice promoted *C. albicans* colonization, which in turn increased inflammation, generating ASCA [14]. Major players of innate immunity sensing *C. albicans*, such as galectin-3 (Gal-3) [14] and toll-like receptor (TLR)2, ref. [15] were also involved in this inflammatory interplay in ICI. 

Considerable progress has been made in our understanding of the complex immunological mechanisms that finely tune the response against *C. albicans* and balance invasion control and inflammatory damage [16,17,18]. It has been established that intestinal inflammation increases the population of *C. albicans* and cross-reactive Th17 cells, highlighting anti-*C. albicans* immunity as a central mechanism for the induction of the human antifungal Th17 response [19]. Furthermore, this systemic pathway extends to heterologous fungal species, such as those involved in pulmonary inflammatory diseases [19,20]. This unforeseen power, developed in the natural *C. albicans* gut biotope, is coherent with many observations suggesting a link between Th17 exacerbation in inflammatory bowel disease (IBD) and *C. albicans* [18,21]. 

Over the past decade, next generation sequencing (NGS) methods of intestinal microbiota analysis have gradually replaced methods of living yeast isolation and identification. Due to the relatively low abundance of fungi among bacteria, ref. [22] focus on the so-called mycobiome [23] has been relatively recent [24]. Interestingly, in all microbiome studies conducted to date in IBD patients, the core mycobiome is dominated by *Candida* species, which might explain why candidiasis [25] is the most common manifestation of microbial dysbiosis [26,27]. These studies suggest, in turn, that a balanced gut mycobiota contributes to the maintenance of host immune homeostasis [28].

Thus, a substantial body of clinical, experimental, mycological, and immunological evidence supports the role of *C. albicans* in triggering or exacerbating CD [28,29]. However, to date, no therapeutic evidence supports this hypothesis. 

We carried out a pilot clinical study to assess the effect of fluconazole (FCZ) [30] on post-operative recurrence of CD. In the absence of any indication for this drug in CD, the trial was limited in size. Due to the low number of patients (*n* = 28) and the use of suboptimal doses of FCZ, the study was not conclusive clinically. However, careful patient follow-up over 6 months with regular mycological and serological sampling provided unique samples to follow the evolution of *C. albicans* colonization together with measurement of the adaptive and innate immunity parameters linked to both *C. albicans* growth and CD inflammation. These biological parameters were scored, and their relevance analyzed individually before multivariate analysis. Our results show that surgical reduction of inflammation and FCZ have an impact on *C. albicans* survival and decrease immune stimulation, while multivariate analysis suggested the influence of fluconazole on biomarkers linked to recurrence.

## 2. Methods

### 2.1. Study Design and Patient Population

This prospective, randomized, double-blind, placebo-controlled study was registered at ClinicalTrials.gov under the ID: NCT02997059, 19 December 2016 [31] where all details of the study are provided. In brief, patients were enrolled between 2009 and 2015 in Lille University Hospital, France. Patients were eligible for the study if they were ≥18-years-old, with ileal or ileo-colic CD, with surgical resection of all macroscopic damage and with a pre-operative ASCA level of ≥50 arbitrary units (AU) (IBDX kit; Glycominds, Lod, Israel) in the month before inclusion. The main exclusion criteria were: pregnancy or no effective contraception, cumulative ileal surgical resection of more than 1 m or subtotal colic resection, hypersensitivity to FCZ or other azoles, and liver or renal failure. The study was approved by the local Ethical Board “Comité Consultatif de Protection des Personnes dans La Recherche Biomédicale de Lille”. Initial approval 15 June 2006 under the number CP 06/59, revised 8 January 2013. Twenty-nine patients (9 males, 20 females; median age: 29-years) were included. The clinical features of these patients are summarized in Table 1. The patients were randomized in two groups: (i) 14 patients were treated with FCZ 200 mg/day; and (ii) 15 patients received a placebo for 6 months (Figure 1). Clinical evaluation of each patient was performed at the time of consent during a pre-operative visit. Patients were then followed-up with 1-, 2-, 3-, and 6-months post-surgery; mycological and blood samples were taken at each visit. Endoscopic evaluation of CD recurrence was performed at 6 months and concerned 21 patients, 12 in the FCZ group and 9 in the placebo group. Study blinding was lifted after collection of all clinical and biological data. Among the 29 patients included initially, 28 had data on colonization and all biological parameters at inclusion and 23 had biological data available at 6 months.

### 2.2. Mycological Analysis

Oral swabs and stools were seeded onto chromogenic agar plates (CandiSelect 4; BioRad^®^, Marnes la Coquette, France) and incubated at 37 °C under standard conditions. The isolated yeasts were identified by MALDI-TOF mass spectrometry (Microflex LT TM Instrument, Bruker Daltonics, Bremen, Germany) as described previously [32]. A colonization score was established as follows: 1 ≤ 10 colony-forming units (cfu); 2 = 10–50 cfu; 3 ≥ 50 cfu; 4 ≥ 100/confluent [10]. Stools samples could not be obtained at some patient visits.

### 2.3. Serological Analysis

Blood samples were separated by centrifugation and serum was kept frozen at −80 °C until use.

All sera were analyzed at the same time for the different biomarkers. The different serum biomarkers studied and the rationale for their inclusion in this study with regard to their relation to both CD and candidiasis are summarized in Table 2.

#### 2.3.1. Serum Markers of Inflammation 

Calprotectin levels were determined with a MRP8/14 sandwich ELISA test (Bühlmann, Cergy le Haut, France) designed on the basis of previous studies; the limit of detection of this kit is 400 ng/mL with median values of 8892 ng and 1318 ng/mL in CD patients and controls, respectively [33]. The test was performed according to the manufacturer’s instructions.

#### 2.3.2. Serological Markers of IBD

Anti-glycan antibodies were detected with the IBDX^®^ panel (Glycominds, Lod, Israel), comprising ASCA, AMCA, ALCA, and ACCA kits involving *S. cerevisiae* mannan, mannobioside, laminaribioside, and chitobioside antigens, respectively [12]. All assays were performed according to the manufacturer’s instructions. The results are expressed as AU relative to the Glycominds laboratory calibrator with positive cut-off values of 50, 100, 60, and 90 AU, respectively.

#### 2.3.3. Markers of Invasive Candidiasis 

Anti-*Candida* mannan antibodies were measured with the PlateliaTM *Candida* Ab Plus ELISA kit (Bio-rad, Marnes la Coquette, France) [34]. The cut-off of this test is 10 AU and the test was designated as CalbManAb.

Anti-*C. albicans* pathogenic phase IgG antibodies were detected by ELISA using recombinant Hyphal wall protein-1 Hwp1 produced in *Escherichia coli*, as described previously [35]. The cut-off value of the anti-Hwp1 Ab test established with cohorts of patients with ICI and appropriate hospital controls is 25 AU. This test was designated as CalbProtAb.

Circulating fungal glucans were detected with the Fungitell asssay (Associates of Cape Cod, East Flamouth MA, USA). This test detects the presence of fungal cell wall *β*-D-glucan (BDG) through a specific biochemical test derived from the Limulus test for endotoxin detection. It is recommended worldwide as a surrogate marker for the diagnosis of ICI and is used routinely in our laboratory [36]. Some papers have reported elevated BDG levels during CD [37,38].

#### 2.3.4. Innate Immunity Lectins Sensing *C. albicans*

Gal-3 is a lectin of innate immunity with a variety of important roles. Its ability to sense mannosides with a *α*-1,2 anomery, rare in the living world but largely expressed on the *C. albicans* surface, was discovered by our group [14,15,39]. Gal-3 concentration was measured with the Qantikine Human Galectin-3-Immunoassay (R&D Systems, Minneapolis, MN, USA) according to manufacturer’s instructions. The normal values are 6.73 ng/mL in healthy volunteers and a cut-off value of 20.6 ng/mL discriminates infectious and non-infectious inflammation with a specificity of 95% [40].

Mannose binding lectin (MBL) is an important lectin of innate immunity triggering many defense mechanisms including the complement system. This lectin senses the ubiquitous *α*-mannosides expressed at high levels on the *S. cerevisiae* and *C. albicans* yeast surface. Levels of MBL were determined with the MBL oligomer ELISA kit (BioPorto Diagnostics, Hellerup, Denmark) involving a monoclonal antibody against the MBL carbohydrate-binding domain, according to manufacturer’s instructions. Despite the absence of consensus, it is generally agreed that MBL concentrations ≤500 ng/mL reflect deficiency and that values <100 ng/mL are associated with increased susceptibility to infection [41,42].

### 2.4. Statistical Analysis 

All statistics and figures were computed with R statistical open-source software, SAS 9.4 (SAS Institute, Cary, NC, USA), and GraphPad Prism v6.0. Software GraphPad Software Inc., San Diego, CA, USA Categorical variables are expressed as number (percentage). Quantitative variables are described as median and interquartile range (IQR). Normality was assessed by the Shapiro-Wilks test combined with graphical evaluation (QQplot).

First, exploratory univariate analyses were performed to assess the evolution of each biomarker after surgery by comparing the value at inclusion with the value at the end of the study (6 months) using the paired Student’s *t* test in case of normality and the non-parametric rank signed test otherwise. These analyses were adjusted for the level at inclusion. In case of non-normality of biological parameters, these analyses were performed on log-transformed data in case of log-normality and on rank data otherwise. Correlations between biological parameters are given by the Pearson or Spearman coefficient of correlation depending on the distribution of the parameters. Considering the exploratory nature of theses analyses, all statistical tests were performed at the two-tailed *α* level of 0.05 and the results should be interpreted as hypothesis-generating. 

Second, statistics and figures were computed with the “R/Bioconductor” statistical open-source software [58]. Biomarkers were described by median and IQR. In order to guarantee normally distributed data, each biomarker data has been transformed using Box–Cox transformation [59]. A differential analysis was carried out using the most appropriate test between paired Wilcoxon signed-rank test or paired student *t*-test according to normality and sample variance distribution. Adjusted *p*-value below 0.05 was considered as statistically significant according to the Benjamini and Hochberg approach [60]. Correlations between biological parameters are given by the Pearson or Spearman coefficient of correlation depending on the distribution of the parameters. The accuracy of each biomarker and its discriminatory power was evaluated using a receiver operating characteristic (ROC) analysis. In addition, all biomarkers were combined with each other to evaluate the potential increase in sensibility and specificity using a linear combination which maximizes the area under the curve (AUC) ROC [61]. The equation for the respective combination is provided and can be used as a new virtual marker Z where *a* is the calculated coefficient and *j* is the biomarker number.as follows:(1)$$Z=\overset{n}{\underset{j=1}{\sum }}a_{j}BMK_{j}$$

Contingency tables have been performed from multivariate mROC biomarkers combination results and chi-squared tests were performed on these tables.

## 3. Results

### 3.1. Qualitative and Quantitative Evolution of Yeast Gut Colonization over the 6-Month Post-Operative Period

The main results are shown in Figure 2. Analysis of oral and fecal yeasts in the 28 patients sampled before surgery showed that 18 patients were colonized (64%): five patients orally, seven fecally, and six in both. The species recovered were *C. albicans* (*n* = 16), *C. tropicalis* (*n* = 3), and *C. lusitaniae* (*n* = 1). Post-operatively, patients were characterized by a gradual reduction in *C. albicans* isolates and a broader spectrum of species isolated, which comprised *C. krusei, C. parapsilosis, C. glabrata, C. lambica, C. norvegiensis,* and *S. cerevisiae* (Figure 2A). Quantitative assessment of the fungal load (Figure 2B) showed that there was a global reduction in colonization, which mainly concerned *C. albicans*, and the gradual appearance of other yeast species, which was more pronounced in patients treated with FCZ (*p* = 0.01).

Whatever the treatment, an analysis of the number of colonized patients showed that the decrease in yeast colonization at 6 months was associated with a transient increase at 1 month in patients who had a recurrence of CD, suggesting that this temporal variation could be critical for the outcome.

Together these results show that, whatever the treatment and endoscopic evolution, an important and gradual decrease in *C. albicans* oral and fecal loads is observed during the 6 months post-surgery.

### 3.2. Evolution of C-Reactive Protein (CRP) and Serum Calprotectin Levels

CRP positive values (>3) did not differ significantly between the initial visit and the 6-month visit (*p* = 0.76).

The change in serum calprotectin levels is shown in Figure 3 from the inclusion visit, before surgery, and during regular follow-up visits. These values are presented according to the treatment (Figure 3A), or the presence of endoscopic recurrence at 6 months (Figure 3B). When all patients are considered, the decrease in serum calprotectin levels was highly significant over the 6-month period (*p* = 0.002). Due to the limited number of patients in each group, the differences were not significantly associated with treatment (*p* = 0.09) or recurrence (*p* = 0.202); however, it is worth noting that the only two obvious increases at the end of the survey corresponded to patients with recurrence of CD.

### 3.3. Evolution of Anti-Glycan Antibody Markers of CD 

Figure 4A–D show the values of ASCA, ALCA, AMCA, and ACCA, respectively, at inclusion and at the last visit (6 months). For ASCA, all patients were ASCA positive (inclusion criterion) and a significant decrease was observed 6 months after surgery (*p* < 0.0001). However, all patients except two remained above the positivity threshold at the end of follow-up. For ACCA, the decreasing trend was less systematic, but significant (*p* = 0.04). No significant trend was observed for ALCA (*p* = 0.11), or AMCA (*p* = 0.33). No association was found between the decrease in levels, treatment, or recurrence. According to the clinical interpretation of the ASCA, ALCA, and ACCA tests, based on the recommended cut-off values, the few patients who evolved from a positive to a negative status (i.e., only two for ASCA) had initial values slightly above the cut-off values. The fluctuations in AMCA were of higher magnitude, decreasing or increasing, leading to more changes from a positive to a negative status. However, consideration of the results from the entire IBD panel would not have affected the diagnosis of CD.

### 3.4. Evolution of Biomarkers of Invasive Candidiasis 

The measurement of anti-*C. albicans* antibodies, produced during tissue invasion by *C. albicans,* is used to survey patients at risk of nosocomial ICI. Our results are shown in Figure 4E,F. Surprisingly, at inclusion, five of the CD patients had significant high titers in these tests. Their anti-*Candida* antibody levels decreased during the survey and reached non-significant levels at 6 months. Anti-*Candida* mannan antibodies (Figure 4E) and anti-Hwp1 antibodies (Figure 4F) exhibited a significant decrease between surgery and 6 months (*p* < 0.0001). As for yeast colonization, no influence other than surgery (i.e., FCZ or recurrence) was significantly related to this decrease. When considering the relationship between the decrease in anti-*C. albicans* antibodies and anti-glycan markers of CD, the decrease in anti-*C. albicans* mannan antibodies tended to be associated with a decrease in ASCA (*p* = 0.069).

Of note, a comparison of BDG levels before surgery and 6 months later gave non-significant results (data not shown).

### 3.5. Level and Evolution of Sensing of C. albicans by the Innate Immunity Lectins Gal-3 and MBL

Gal-3 is a cell membrane-associated pattern recognition receptor binding *C. albicans* surface mannosides. As shown in Figure 4G, most patients had higher Gal-3 levels than those observed in healthy subjects (>4 ng/mL). A significant decrease was observed 6 months after intestinal resection (*p* < 0.0001). As for anti-*C. albicans* antibodies and yeast colonization, no influence other than surgery could be established for this decrease. Gal-3 and calprotectin levels correlated before surgery and at the end of the survey (*p* = 0.05 and *p* = 0.03, respectively). During the survey, the decreases in serum Gal-3 and calprotectin were correlated (*p* = 0.0001). MBL is another soluble lectin playing an important role in innate immunity with regard to *C. albicans*. It senses the massive bulk of cell wall mannan surface α-mannosides. Almost half of CD patients (*n* = 11, 48%) had low or very low MBL levels (<1000) at inclusion. As shown in Figure 4H, MBL levels remained relatively stable between inclusion and the last visit (*p* = 0.283). When the correlation between MBL levels and other biomarkers was analyzed, a significant relationship was found between low MBL levels and high levels of anti-*C. albicans* antibodies before surgery (correlation of −0.39; *p* = 0.04), suggesting a compensation by adaptive immunity for the innate MBL defect. After surgery and a decrease in *Candida* colonization this relation disappeared (correlation of −0.18; *p* = 0.40). The results from the analysis of the ability of combined *Candida* and CD biomarkers to discriminate between surgery and 6 months later are summarized in Table 3.

### 3.6. Integration of the Biomarkers Tested in Diagnostic and Prognostic Models

Considering that the biomarkers related to *Candida* were affected by clinical CD remission after surgery, we first analyzed whether one biomarker, or a combination, could discriminate patient status before and 6 months after surgery using multivariate ROC curves. As a reference marker of systemic inflammation, we used calprotectin which provided acceptable results in terms of specificity and sensitivity (AUC = 0.76) (Figure 5). However, this performance was similar to that of Gal-3, which was included in this study as a marker of CD and an innate immunity receptor for *C. albicans* (AUC = 0.76). When combined with anti-*C. albicans* antibodies the AUC was an impressive 0.86 suggesting that, as for colonization, a cause-to-effect relationship exists in relation to the reduction of inflammation by surgery. Of note, although BDG values analyzed individually provided non-significant results, their combination with anti-*C. albicans* mannan antibodies and Gal-3 gave an impressive AUC = 0.84 supporting the impact of CD surgery on *C. albicans* biomarkers.

We then analyze whether some factors could be predictive of relapses over time. In this respect, we analyzed the evolution of different combinations of markers at each visit. As shown in Figure 6A, different combination of anti-glycan antibodies, anti-*C. albicans* antibodies, circulating BDG, and calprotectin could predict relapses (the threshold is represented by a dotted line). Interestingly, patients were not “equal” in this analysis since biomarker levels already differed at the beginning of the survey; the most discriminating time in the evolution was 2 months.

The next analysis concerned the effect of antifungal treatment. In contrast to the previous analysis, which reflected on adaptive immunity biomarkers as predictors of relapse, the innate immunity lectins sensing *C. albicans,* Gal-3 and MBL, appeared more determinant for sensing a potential role of antifungals on disease evolution (Figure 6B). Although these biomarkers overlapped before surgery, their profiles diverged clearly 1 month later, having a maximum difference according to antifungal treatment at 2 and 3 months.

The final analysis concerned the effect of FCZ on CD recurrence by combining the two previous analyses (occurrence of relapses and effect of FCZ) at each visit (Table 4).

When patients who were positive for the Gal-3 + MBL + calprotectin combination were considered (i.e., the true positives (patients responding to FCZ treatment) and the true negatives (negative test in placebo patients)) there was a significant difference in terms of recurrence between the two populations (Chi^2^ test; *p* = 0.007). The number of recurrences appeared to be lower in patients treated with FCZ compared to placebo.

When considering patients who respond positively to an association of adaptive immunity biomarkers with lectins (Figure 4), the combination ACCA + Platelia Ab + MBL + calprotectin (i.e., the true positives (patients responding to FCZ treatment) and the true negatives (negative test in placebo patients)), a significant difference in terms of recurrence was observed between the two populations (Chi^2^ test; *p* = 0.016). Thus, the role of adaptive immunity was less noticeable than that of innate immunity regarding relapses in relation to FCZ.

Altogether it appears that biomarkers of adaptive or innate immunity targeting *C. albicans* could have an evolution that is predictive of CD recurrence, and that antifungals with the same target (*C. albicans*) also have an influence on recurrence.

## 4. Discussion

This study describes the high rate of *C. albicans* colonization in CD patients, with 18/23 (78%) patients sampled before surgery colonized. This is in accordance with several clinical [10] and experimental [14,18] studies showing that survival of *C. albicans* in the gut is favored by inflammation. *C. albicans* was the predominant species isolated, sometimes associated with *C. tropicalis. C. tropicalis* has been described in some NGS mycobiota analyses [27,51] as a predominant yeast species in CD in contrast to other studies, ref. [26] which clearly demonstrated *C. albicans*. Of note, during this survey involving more than 200 samples, *S. cerevisiae* was only isolated once, in contrast to a study that attributed an important deleterious role for this species on CD evolution [62,63]. Globally, this longitudinal study shows that, following CD surgical resection, and independently of treatment, both the number of colonized subjects and fungal load decline, the mycobiome became less monomorphic with the appearance of species with reduced adaption to endosaprophytic growth, or presenting intrinsic or acquired resistance to FCZ [64]. This is the first evidence that decreasing inflammation by intestinal resection decreases *C. albicans* colonization. Interestingly, this decrease in colonization was also observed in the oral cavity, suggesting a systemic effect.

The clinical conclusions of this pilot study are restricted by the limited number of CD patients and the absence of endoscopic documentation in some of them. Only a very slight trend towards a higher proportion of recurrences in the placebo group was observed. In contrast to clinical analysis, for which the study was obviously underpowered, patient and medical team investment during the study led to the collection of an impressive number of samples. We therefore focused on a biological analysis of the host-*Candida* interplay to assess whether some biomarkers could reflect disease evolution.

Fecal calprotectin is a recognized biomarker of IBD [44,65]. Some clinical [33] and experimental [43] studies have suggested that measurement of serum calprotectin could also be useful in this context. In our study, serum calprotectin levels dropped significantly at 6 months post-surgery; only two increases were observed at this date, both in patients with CD recurrence.

ASCA were the first antibodies directed against yeast glycans to be associated with CD [3]. After two decades and more than 400 studies, ASCA are recognized as the most potent serological marker of CD [3,6,8]. Further studies by Dotan et al. demonstrated that a panel of synthetic glycans detecting antibodies designated AMCA, ALCA, and ACCA complement ASCA as serological biomarkers of CD [10]. In this panel, named IBDX (Glycominds), the number of biomarkers and magnitude of the antibody response is predictive of CD severity [47,66]. Several papers have reported that these antibodies are stable over a patient’s lifetime independently of surgical or medical treatments [67,68]. Clinical studies focusing on immunogens for ALCA and ACCA have shown that, as well as ASCA, these antibodies can be generated during *C. albicans* infection [13]. We observed, over a short time period after intestinal resection, a significant decrease in these antibodies, which was limited in magnitude since it did not change the CD status of the patients. Such a transient post-operative decrease has already been reported and was attributed to possible interference with C. *albicans* stimulation [69]. In the present study, we assessed, in parallel to IBDX, the level of antibodies used as markers of *C. albicans* infection (i.e., Platelia Ab and anti-Hwp1 Ab). Before surgery, these antibodies were present in up to 16% of patients at levels not found in the control population [34,35,70]. This unusual prevalence could reflect dysregulation of *C. albicans* tolerance, or conditions promoting the virulence of this species. It is worth noting that surgery and the associated reduction in *Candida* colonization led, in all patients, to these antibodies falling to levels not discriminating colonization from infection. It is therefore possible that this decrease in anti-*C. albicans* antibodies may account, at least in part, for the decrease in anti-glycan antibodies known to be generated by the pathogenic phase of *C. albicans* [13,49]. The significant correlations found between a decrease in anti-*C. albicans* antibodies and the IBDX panel seem to support this hypothesis, which could result from reduction in *C. albicans* burden.

The rationale for exploring Gal-3 levels in this study was twofold. First, Gal-3 has been identified by our group as the receptor for *C. albicans α*-1,2 oligomannosides acting as immunomodulatory adhesins [39]. This interaction has been confirmed as determinant for enhancing TNF-*α* production after cross-talk with TLR-2 [15] and dectin-1 [52]. Second, increased Gal-3 levels have been associated with CD, ref. [53,71] including triggering the activity of fibroblasts which are important players in the evolution of CD [72,73]. It is therefore not surprising that Gal-3 levels decrease following a reduction in inflammation by surgery, which in turn reduces the *C. albicans* gut burden. In a previous study, Netea et al. showed that higher levels of Gal-3 were associated more with *Candida* infection than with inflammatory diseases; however, CD was not among the inflammatory diseases studied [40]. Some observational clues from this study suggest that it would be worthwhile to focus research on possible amplification loops between Gal-3, *C. albicans* and gut inflammation/fibrosis although the expression of *C. albicans α*-mannosides, reduced in vivo, may have a conflicting role [74].

Like Gal-3, MBL is a lectin playing a dual role in the *C. albicans*-CD interplay [56]. The prominent expression of *α*-mannoside residues at the *C. albicans* cell wall surface led to early interest in this lectin being able to activate the complement system after binding to the yeast and mediate opsonization for clearance; this mechanism is more effective for *Candida* than for bacteria [75]. Several studies have shown a relationship between MBL defects and increased susceptibility to *C. albicans* infection [76,77]. With regards to CD, a high proportion of patients have MBL deficiency [55,57]. It has been suggested that low levels of MBL could be responsible for promoting antibody responses with cross-reactive potential against common mannan epitopes (ASCA) [56]. Although this observation is still controversial, it has recently been reported that MBL-deficient expression in the mouse gut resulted in increased susceptibility to disseminated infection as well as over-expression of pro-inflammatory interleukins (IL)17 and 23 [54]. It is noteworthy that in the present cohort selected on the basis of significant ASCA levels, the proportion of patients with low MBL levels (<1000) was extremely high (48%). In contrast to Gal-3, MBL levels remained relatively stable during our survey. However, the correlation between low MBL levels and anti-*C. albicans* antibodies observed before surgery disappeared after 6 months, suggesting the absence of a need for compensatory mechanisms when the high gut burden of *C. albicans* is reduced.

Multivariate analysis showed that biomarkers of adaptive or innate immunity targeting *C. albicans* could have an evolution that, in parallel with the reduction in colonization, accounts for differences in patient status before and after surgery. Furthermore, we showed that biomarkers of adaptive or innate immunity targeting *C. albicans* could have an evolution that is predictive of CD recurrence. As far as differences between patients, with recurrence or not, were visible early in follow up (Figure 6A), this would mean that application of the same equation to the biomarkers studied could represent a predictive model that is useful for clinical management. Despite the FCZ dose of 200 mg per day, which is at the lower end of what is needed to obtain meaningful tissue concentrations, ref. [30] we were able to show that FCZ with *C. albicans* as the target also had an influence on CD recurrence.

Interestingly, this conclusion about an effect of FCZ on immune functions via *C. albicans* targeting is coherent with a previous study, which, although not dealing with IBDs but with the prevention of ICI in blood marrow transplant recipients, unexpectedly showed that FCZ was associated with an unexpected reduction in digestive graft-versus-host disease (GvHD) [78]. This observation, which suggested a role for *C. albicans* in T-cell activation, was confirmed by another study that showed a role for *C. albicans* in GvHD through its interaction with dectin-1 [78]. Despite these observations, the role of FCZ on inflammation through its activity on *C. albicans* has never been investigated in IBDs.

Basic research on *C. albicans* pathogenic traits has reached a very a high standard over the past few decades following the declaration of *C. albicans* as a major human pathogen due to improvements in medical practices. As this increased incidence in ICI was concomitant with the increased incidence of IBD, it is surprising that few mycologists have investigated the role of *C. albicans* in IBDs despite information gained from analysis of the mycobiota. This first translational study on the evolution of myco-immunological parameters associated with CD after surgical resection highlights a strong relationship between *C. albicans* survival, its immune control, and inflammation.

From a mycological point of view, this is the first study to report that decreasing inflammation by intestinal resection decreases *C. albicans* colonization-living load. This provides a logical counterargument to the numerous observations that inflammation increases *C. albicans* colonization, which in turn increases inflammation. Interestingly, the biomarkers studied, either related to *C. albicans* pathogenicity, or related to both ICI and CD, also decreased. Finally, building multivariate biostatistical models has established significant links between CD recurrence, immune sensing of *C. albicans*, and the positive effects of antifungal treatment. This fulfilled the aims of this pilot study, namely to define how an anti-*C. albicans* treatment could interfere with CD evolution. The results suggest that prospective studies intended to assess the effect of new generations of antifungal drugs on CD evolution would be worthwhile. More generally, cumulative data on the deleterious role of *C. albicans* in CD also raise the question of the value of an anti-*Candida* vaccine strategy to limit these effects.

## Figures and Tables

**Figure 1 jof-07-00324-f001:**
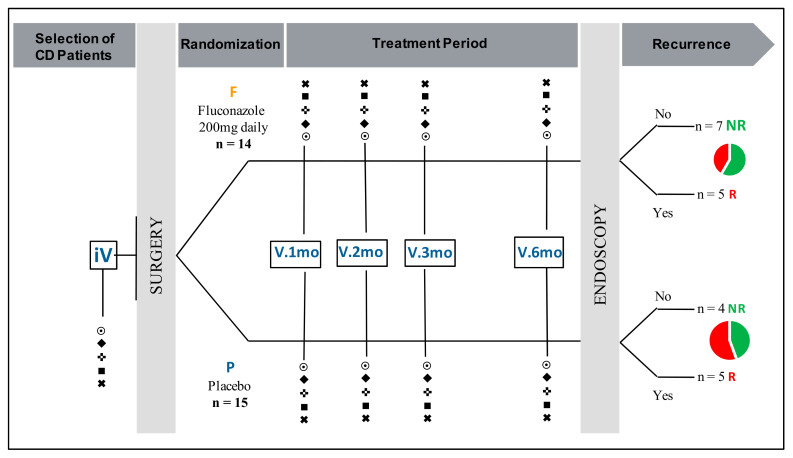
Study design and patient population. **iV**: initial visit, **V.1mo**, **V.2mo**, **V.3mo**, **V.6mo**: visits at 1, 2, 3, and 6 months. **P**: placebo; **F**: fluconazole; **NR**: no recurrence; **R**: recurrence. ⨀ Mouth swabs and stool sampling: isolation, identification, and quantification of yeasts. Serum sampling: (♦) calprotectin levels; (✜) Crohn’s disease (CD) anti-glycan antibodies (ASCA, anti-chitobioside (ACCA), anti-laminaribioside (ALCA), and AMCA); (■) anti-*C. albicans* antibodies (CalbManAb, CalbProtAb); (✖) innate immunity lectins: galectin-3, mannose binding lectin.

**Figure 2 jof-07-00324-f002:**
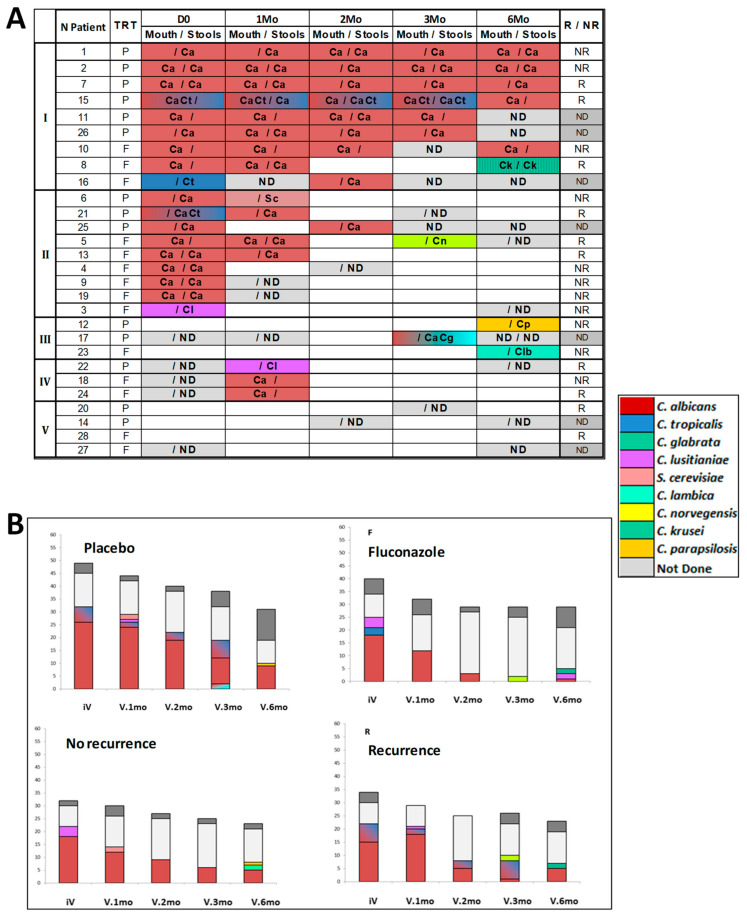
Evolution of yeast colonization. (**A**) qualitative evolution of digestive tract (mouth and stools) yeast colonization. Group I: patients initially colonized and remaining colonized during the survey period (*n* = 9). Group II: patients initially colonized with mycological “sterilization” (*n* = 9) after surgery. Groups III and IV: patients with no yeast colonization at inclusion, with yeasts isolated transiently at V.2Mo (*n* = 3), or V.6mo, last visit (*n* = 3). Group V: patients never colonized (*n* = 4). The yeast species isolated are represented by initials and color codes as specified on the Figure. P: placebo; F: FCZ; NR: no endoscopic recurrence; R: endoscopic recurrence at V. 6mo; NP: endoscopy not performed. Each species is represented by color code as specified on insert, initials of genera and species names are additionally reported on the graph, when 2 species are isolated ex CaCt colors are mixed. (**B**) quantitative evolution of yeast colonization in the mouth and stools according to treatment with fluconazole (FCZ), or placebo (P), and evolution towards no endoscopic recurrence (NR), or endoscopic recurrence (R). Quantitative assessment was carried out by determining the colonization score (see Patients and Methods) for species isolated at each visit and represented by color codes (as in (**A**)).

**Figure 3 jof-07-00324-f003:**
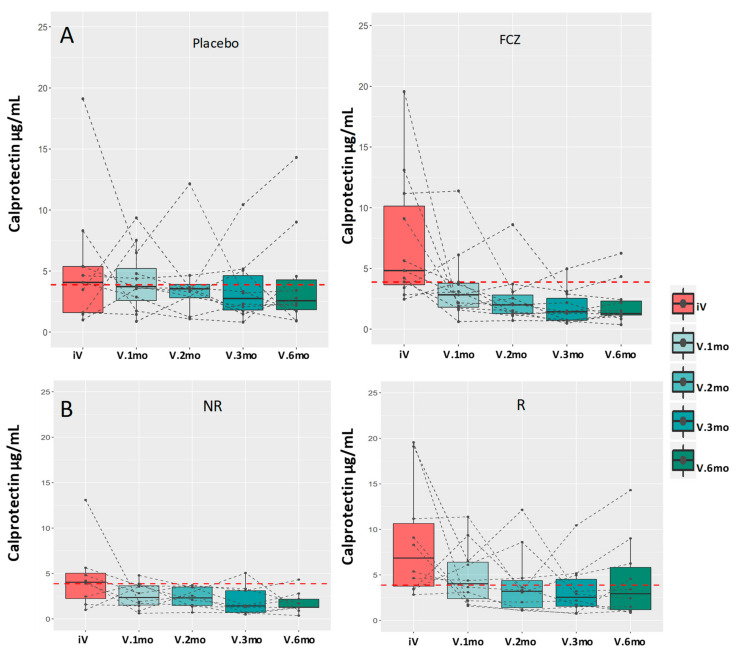
Comparative kinetic evolution of serum calprotectin levels (μg/mL) after surgery (**Panel A**): in patients treated with placebo (**P**), or FCZ (**F**), and (**Panel B**); in patients with no recurrence (**NR**) or recurrence (**R**).

**Figure 4 jof-07-00324-f004:**
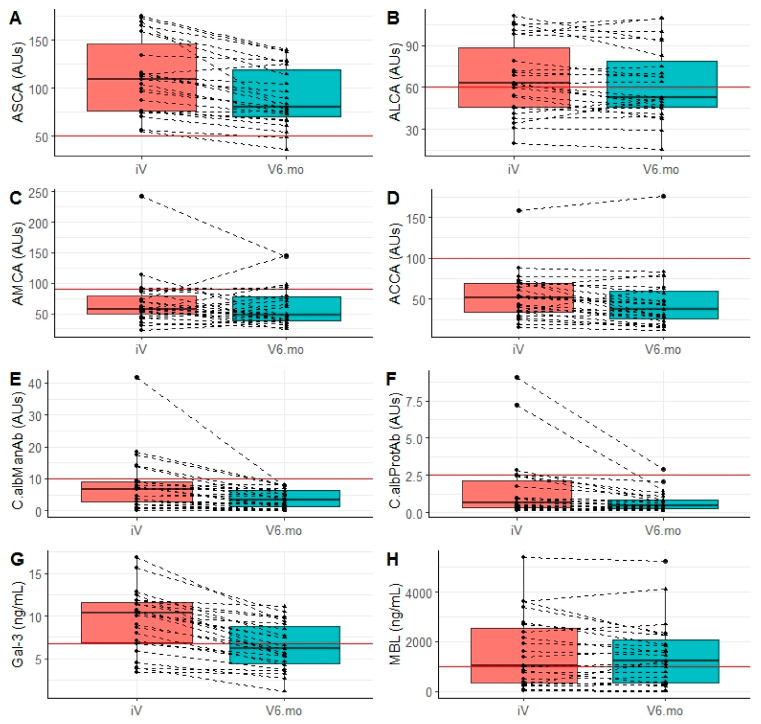
Evolution of biomarkers of CD and *C. albicans* colonization before surgery and 6 months post-surgery. (**A**–**D**) anti-glycan antibodies used for CD diagnosis and stratification: (**A**) ASCA, (**B**) ALCA, (**C**) AMCA, and (**D**) ACCA. Results are expressed in arbitrary units (AUs), and cut-offs correspond to the red line. (**E**,**F**) Antibody markers of *C. albicans* pathogenic development: (**E**) PlateliaAb^®^ test (C.albManAb) and (**F**) anti-Hwp1 (C.albProtAb). Results are expressed in AUs, cut-offs correspond to the red line. (**G**,**H**) innate immunity lectins sensing *C. albicans* mannose residues: (**G**) galectin-3 (Gal-3), sensing surface *α*-mannosides, and (**H**) mannose binding lectin (MBL), sensing *α*-mannosides. Results are expressed in AUs, Gal-3 and MBL in ng/mL cut-offs correspond to the red line.

**Figure 5 jof-07-00324-f005:**
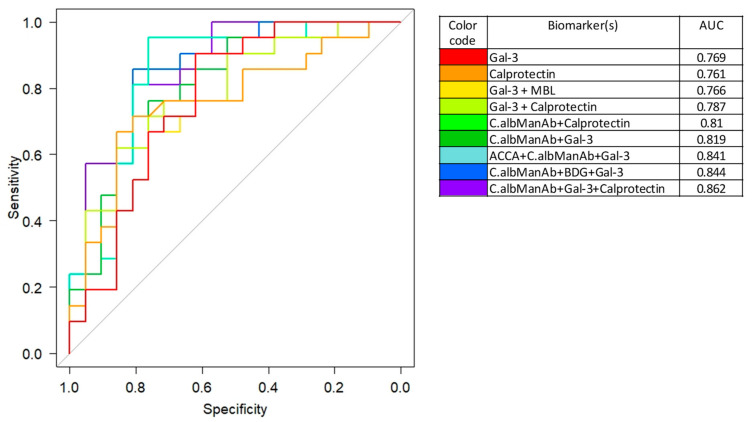
Performance of the different biomarkers for discriminating patient parameters before surgery and 6 months post-surgery. Results are expressed graphically by receiver operating characteristic (ROC) curves and by the corresponding area under the curve (AUC) values (right of the figure with color codes). With reference to calprotectin alone as a systemic marker of inflammation, galectin-3 performed just as well in this clinical setting. When combined (mROC) with *C. albicans* biomarkers, impressive AUCs were reached showing an elective impact of CD surgery on *C. albicans* survival/sensing by the host.

**Figure 6 jof-07-00324-f006:**
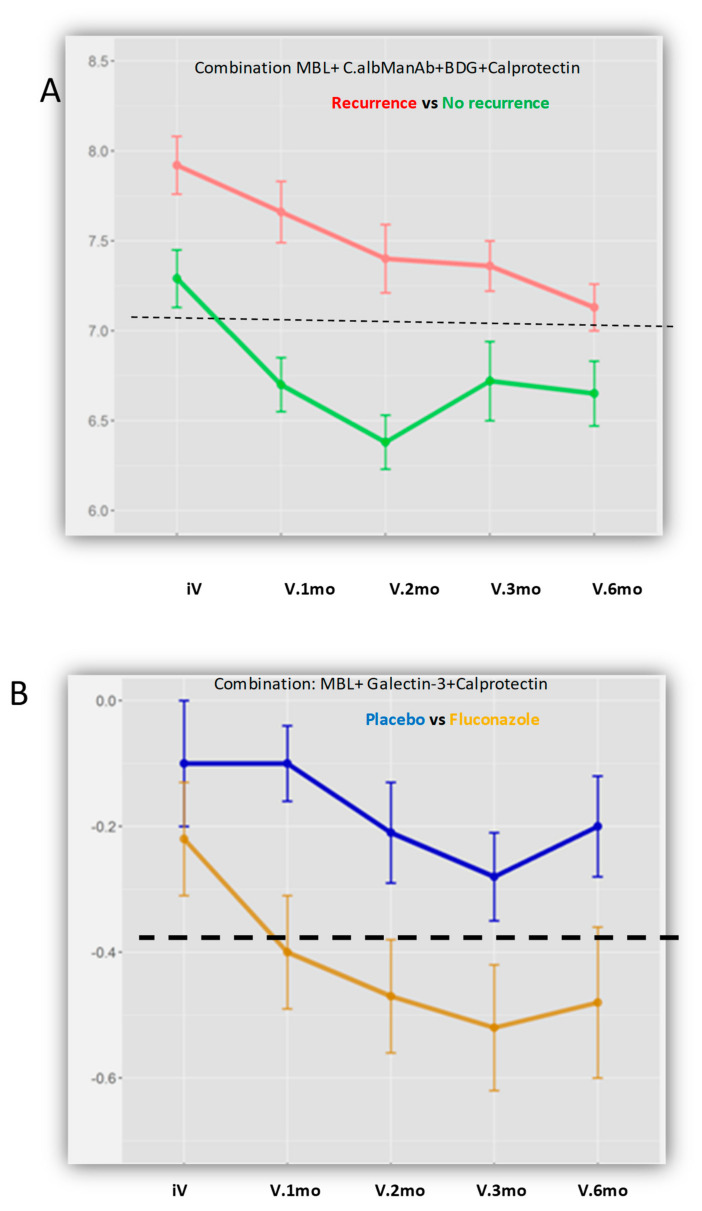
Evolution of different combinations of markers at each visit. (**A**) for the prediction of recurrence. Example of the combination MBL + calprotectin + C.albManAb + *β*-D-glucan (BDG), which could predict recurrence with reference to the threshold represented by the dotted line. In this figure, the patients differed before surgery while major differences occurred 2 months later. (**B**) according to the effect of FCZ treatment. Example with MBL + galectin-3 + calprotectin, which reflects the effect of FCZ on CD evolution. Although the biomarkers overlapped before surgery, their profiles diverged clearly 1 month later, with a maximum difference at 2 and 3 months. Of note, the more discriminant combinations always included calprotectin, reflecting inflammation levels, but not antifungal treatment.

**Table 1 jof-07-00324-t001:** Characteristics of the patients included in the placebo and antifungal treatment groups, after lifting of blinding.

	FCZ Group (*n* = 14)	Placebo Group (*n* = 15)
Sex (male)	5 (35.7)	4 (26.7)
Age at CD diagnosis (years)	20.3 (19.4–21.8)	23.4 (20.5–31.7)
Age at inclusion	26.7 (22.8–34.2)	29.5 (23.6–40.8)
Disease duration (years)	4.9 (1.9–14.1)	3.9 (0.5–9.1)
Disease type		
Penetrating	4 (28.6)	6 (40.0)
Stenosing	9 (64.3)	9 (60.0)
Unclassified	1 (7.1)	0 (0.0)
Previous surgery	0	0
ASCA titer at inclusion	86.5 (F70.3–114.8]	108.4 (78.4–134.3)
CDAI at inclusion	135 (86–217)	104 (68–127)

All values shown are *n* (%), or median (interquartile range). FCZ: fluconazole; ASCA: anti-*Saccharomyces cerevisiae* antibodies; CDAI: Crohn’s disease activity index.

**Table 2 jof-07-00324-t002:** List of biomarkers analyzed and rationale for their inclusion in this study.

Type of Serum Biomarker Analyzed/*Abbreviation*	General Features	Molecular Characteristics	Rationale for Longitudinal Analysis in Relation with Crohn’s Disease Evolution	Rationale for Longitudinal Analysis in Relation with *Candida* Colonization/Infection
**Markers of inflammation**				
**Calprotectin**	Described as biomarker of systemic inflammation [33,43]. Detection of feacal calprotectin in stools is recommended for IBD diagnosis and treatment follow up [44,45]	Protein from PMNs (S100 A8/A9 Family)	High levels suggested as being informative for diagnosis and disease outcome [33,43,44]	None current indication for the diagnosis of Candidiasis.
**Serum Antibodies**				
***ASCA***	Anti-*Saccharomyces cerevisiae* antibodies	Anti-oligomannose antibodies (Epitope present in *S. cerevisiae* and *C. albicans* mannan) Epitope with Man *α*-1,3 linked at the non reducing end of tetra or trimannose *α*-1,2 linked [3].	The most potent marker of Crohn’s disease in terms of prevalence and prediction of onset [8,9]	Generated during a *C. albicans* invasive infection [13] In CD patients healthy relatives their presence correlates with *C. albicans* carriage [10].
***ALCA***	Anti-Laminaribioside Carbohydrate Antibodies	Disaccharide sequence of *α*-1,3 glucans(Epitope present in *S. cerevisiae* and *C. albicans* glucans)	Adjunct of ASCA for CD diagnosis [12]	Generated during a *C. albicans* invasive infection [13] Antibodies targeting molecules interacting with Dectin-1, a major receptor of *C. albicans* innate immunity triggering IL17 pathway [46]
***ACCA***	Anti-Chitobioside Carbohydrate Antibodies	Disaccharide sequence of chitin (Epitope present in *S. cerevisiae* and *C. albicans* chitin)	Adjunct of ASCA for CD diagnosis [12,47]	Generated during a *C. albicans* invasive infection [13] The target of these antibodies (chitin) binding to FIBCD1 dampen Intestinal inflammation [48]
**Platelia test** **^®^** ***C.albManAb***	Anti-Candida albicans mannan Antibodies	Detection of human polyclonal antibody response to mannan oligomannose repertoire	Levels increased [10,49]	Marker of *C. albicans* invasive infection used for diagnosis [13,34]
**Anti-Hwp1 Antibodies/*C.albProtAb***	Anti-Hyphal Protein 1 antibody Hwp1 is a molecule specific for *C. albicans* hyphal (invasive) form [50]	Peptidic epitope [35]	Never explored in this setting	Marker of *C. albicans* invasive infection used for diagnosis [35]
**Fungitel Assay(TM)** ***/BDG***	Circulating *β*-1,3 D Glucan [36]	Polymer of *β*-1,3 D glucan Recognized by Dectin-1 activating the Th17 pathway [29,51]	Described as circulating during CD [37]	Widely used for the diagnosis of systemic candidiasis [36]
**Innate Immunity Lectins**				
**Galectin-3/*Gal-3***	C-Lectin member of Galactose binding lectins Expressed on cells of the macrophage lineage [39]. Expressed on intestinal cells. Associates with dectin-1 for Fungal recognition [52] associates with TLR2 [15].	Receptor for *C. albicans β*-1,2 mannoses, immunomodulatory adhesins, expressed on *C. albicans* mannan mannoproteins and glycolipids [14,15]	Increased during inflammatory diseases [40,53]	Increased during Candida infection [40]
**Mannose Binding Lectin *MBL***	Produced by hepatic and intestinal cells [54] Combines with MASP for activating complement lectin pathway	Reacts with *α*-D-mannose N-acetyl-D Glucosamine L-Fucose	Role in Crohn’s disease: Impaired MBL-MASP functional activity in CD [55] Levels inversely correlates with ASCA [56,57]	Role in mucosal and systemic Candidiasis, participates to clearance of circulating *C. albicans* mannan in patient sera [41,42,54]

**Table 3 jof-07-00324-t003:** Mean values of *C. albicans* and CD biomarkers before surgery and 6 months post-surgery, and significance of their evolution during this period.

Biological Parameter	Visit	All Patients (*n* = 23)	*p* iV vs. V6.mo
Calprotectin	iV	4.2 (3.3–8.7)	0.002
	V6.mo	2.2 (1.2–3.4)	
ASCA	iV	109.2 (76.7–159.0)	<0.0001
	V6.mo	80.6 (67.4–125.1)	
ALCA	iV	62.6 (45.3–97.7)	0.113
	V6.mo	53.0 (45.0–82.8)	
ACCA	iV	51.6 (34.3–69.7)	0.04
	V6.mo	37.6 (23.2–63.0)	
AMCA	iV	57.3 (45.1–85.0)	0.327
	V6.mo	47.4 (38.9–79.8)	
C.albManAb	iV	6.8 (2.7–9.5)	<0.0001
	V6.mo	3.3 (1.0–6.3)	
C.albProtAb	iV	0.6 (0.3–2.4)	<0.0001
	V6.mo	0.4 (0.3–0.8)	
Galectin-3	iV	10.3 (6.7–11.8)	<0.0001
	V6.mo	6.2 (4.2–9.1)	
MBL	iV	1032.4 (271.5–2697.2)	0.283
	V6.mo	1216.0 (304.9–2271.4)	

Values shown are median (interquartile range).

**Table 4 jof-07-00324-t004:** Analysis of the response to treatment: number of post-operative recurrences of CD according to treatment (FCZ) vs. placebo, as analyzed by combinations of biomarkers.

	No Recurrence	Recurrence	Total	*p*(χ^2^)
**Gal-3 + MBL + Calprotectin**				
**Response to treatment**				
**FCZ (+)**	**27**	12	39	
**Placebo (−)**	14	**22**	36	
**Total**	41	34	75	**0.007**
**No response to treatment**				
**FCZ (−)**	7	13		20
**Placebo (+)**	4	3		7
**Total**	11	16		**27**
**Overall total**	52	50		**102**
**ACCA + CaManAb + MBL + Calprotectin**				
**Response to treatment**				
**FCZ (+)**	**24**	9	33	
**Placebo (−)**	14	**23**	37	
**Total**	38	32	70	**0.016**
**No response to treatment**				
**FCZ (−)**	10	16	26	
**Placebo (+)**	4	2	6	
**Total**	14	18	32	
**Overall total**	52	50	**102**	

## Data Availability

The raw data supporting research are available on request from the corresponding author.

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
