# Peer review of "A Pilot Clinical Study on Post-Operative Recurrence Provides Biological Clues for a Role of Candida Yeasts and Fluconazole in Crohn’s Disease"

_jof, 2021, doi:10.3390/jof7050324_

Round 1

Reviewer 1 Report

This is an interesting and potentially very relevant study that assesses for the first time in a clinical setting a hypothesis proposed several years ago by the authors and other researchers, namely that fungal colonization may play an important role in Crohn’s disease (CD). While the study is too small to be able to provide clinical conclusions, the bioinformatic models built on basis of the data accumulated suggests an interaction between Candida albicans and CD.

Comments:

  1. In the Methods it is not clearly described which are the primary endpoints, and which are the secondary and/or exploratory endpoints of the trial.
  2. The decrease in overall colonization with Candida after surgery, even at the level of the oral mucosa, is very interesting. Can the authors speculate regarding the mechanisms?
  3. Do the authors consider colon or oral colonization with Candida albicans equally relevant for CD? Which could be the mechanisms through which oral colonization may impact CD?

Reviewer 2 Report

This prospective, pilot clinical study is aimed at evaluating the effect of low dose fluconazole treatment on post-operative recurrence of Crohn's disease. Although, the the study was inconclusive in terms of the effect of fluconazole as a treatment option, several parallel measurements presented should be of value to the scientific literature. Overall, the manuscript is clearly written, and the results justify the conclusions. Addressing the following concerns may improve the quality of the manuscript:

  1. The title may be misleading particularly on the effect of fluconazole. Similarly the statement "FCZ had a positive against recurrence" in the abstract is not justified by the results.
  2. What is an 'integrative bioinformatic model' mentioned in the abstract? Again, I find this misleading. If this refers to Fig. 5 (ROC) and Fig. 6, these are not really bioinformatic models but statistical tools. This needs be walked back.  
  3. A similar concern arises for the statement in the Introduction (line 120). 
  4. Was a power analysis done to justify the small number of subjects? Can the authors explain what the rationale for 'n', and what were the expectations?
  5. Please label all axes as to what they are (units etc.).
  6. Fig. 5: What is the ROC for Gal-3 + Calprotectin? This could be interesting because these two markers may be broader than for yeasts.
  7. It may help if Table 4 be explained more clearly for non-statisticians (may be in the Methods section)
